# Propyl Gallate Attenuates Methylglyoxal-Induced Alzheimer-like Cognitive Deficits and Neuroinflammation in Mice

**DOI:** 10.3390/ijms27010511

**Published:** 2026-01-04

**Authors:** Hui-Yun Tsai, Jing Qiu, Han-Wei Liao, Chi-I Chang, Yu-Hsiang Chen, Chi-Tang Ho, Yu-Kuo Chen

**Affiliations:** 1Department of Nutrition and Health Science, Fooyin University, Kaohsiung 831301, Taiwan; rhytsai@gmail.com; 2Aging and Disease Prevention Research Center, Fooyin University, Kaohsiung 831301, Taiwan; 3Department of Food Science, National Pingtung University of Science and Technology, Pingtung 912301, Taiwan; asus0979694098@gmail.com (J.Q.); b123599272@gmail.com (H.-W.L.); 4Department of Biological Science and Technology, National Pingtung University of Science and Technology, Pingtung 912301, Taiwan; changchii@mail.npust.edu.tw; 5Department of Medical & Molecular Genetics, Indiana University School of Medicine, Indianapolis, IN 46202, USA; yuhschen@iu.edu; 6Department of Food Science, Rutgers, The State University of New Jersey, New Brunswick, NJ 08901, USA; ctho@sebs.rutgers.edu

**Keywords:** propyl gallate, diabetes, methylglyoxal, Alzheimer-like cognitive impairment, PI3K/Akt/GSK-3β pathway, tau phosphorylation, public health

## Abstract

Methylglyoxal (MG), a reactive dicarbonyl metabolite associated with diabetes and metabolic disorders, contributes to carbonyl stress, neuroinflammation, and Alzheimer-like neurodegeneration. This study investigated the neuroprotective effects of propyl gallate (PG), a phenolic antioxidant widely used as a food additive, against MG-induced cognitive impairment in mice. Male C57BL/6J mice were exposed to 1% MG in drinking water for eight weeks and orally administered PG (20, 40, or 100 mg/kg/d). Behavioral tests demonstrated that PG significantly improved spatial learning and recognition memory and alleviated anxiety-like behavior induced by MG. Histological and biochemical analyses revealed that PG reduced hippocampal neuronal damage, suppressed tau hyperphosphorylation and amyloid-β (Aβ) accumulation, and attenuated the overexpression of pro-inflammatory cytokines TNF-α and IL-6. Furthermore, PG increased PI3K expression and Akt phosphorylation while reducing activation of GSK-3β, counteracting the MG-induced suppression of this pathway and aligning with reduced tau hyperphosphorylation. These findings indicate that PG protects against MG-related cognitive dysfunction through modulation of neuroinflammatory responses and survival-related signaling pathways, highlighting its potential as a neuroprotective dietary antioxidant for metabolic stress-associated neurodegenerative disorders.

## 1. Introduction

Alzheimer’s disease (AD) is the most prevalent neurodegenerative disorder worldwide and a major cause of dementia among the elderly population [1]. Increasing evidence suggests a close association between AD and metabolic disorders such as diabetes mellitus [2,3,4], which share common pathogenic features including insulin resistance, chronic inflammation, oxidative stress, and the overactivation of glycogen synthase kinase-3β (GSK-3β) [5,6,7]. Excessive production of advanced glycation end-products (AGEs) and impairment of glucose metabolism further exacerbate neuronal dysfunction and cognitive decline. Because of these overlapping molecular mechanisms, AD has been conceptualized as “type 3 diabetes mellitus (T3DM) [8],” highlighting the crucial role of metabolic stress and carbonyl toxicity in its pathogenesis.

The hallmark pathological features of AD include extracellular β-amyloid (Aβ) plaques and intracellular neurofibrillary tangles (NFTs) composed of hyperphosphorylated tau protein [9,10,11]. In Alzheimer’s disease, Aβ accumulation results from aberrant enzymatic cleavage of amyloid precursor protein (APP) mediated by β- and γ-secretases. In parallel, excessive activation of tau-related kinases, particularly GSK-3β and cyclin-dependent kinase 5 (CDK5), contributes to abnormal tau phosphorylation and neurofibrillary pathology [12,13]. These aggregated proteins disrupt cytoskeletal stability, impair synaptic signaling, and ultimately lead to neuronal loss and memory impairment [14]. Recent studies suggest that metabolic dysregulation and glycation stress can accelerate both Aβ deposition and tau pathology, bridging glucose metabolism dysfunction and neurodegenerative progression [15,16].

Methylglyoxal (MG) is a reactive α-oxoaldehyde generated as a by-product of glycolysis, lipid peroxidation, and amino acid metabolism [17]. MG readily reacts with proteins, lipids, and nucleic acids to form AGEs, leading to carbonyl stress and cellular injury [18]. Under physiological conditions, MG is detoxified through the glyoxalase system, composed of glyoxalase I (GLO1) and glyoxalase II (GLO2), which use glutathione as a cofactor [19]. Reduced glutathione levels impair MG detoxification, promoting AGE accumulation, oxidative stress, and inflammation [20]. Elevated MG concentrations have been reported in both diabetic and AD patients [21,22], and experimental MG administration induces tau hyperphosphorylation, Aβ accumulation, and memory deficits in wild-type rodents, supporting its validity as a non-transgenic Alzheimer-like model [23,24]. Thus, MG-induced glycation stress provides a suitable platform to evaluate neuroprotective interventions.

Propyl gallate (PG), a synthetic phenolic ester of gallic acid, is widely used as an antioxidant food additive [25]. Beyond its antioxidant function, PG and other phenolic compounds can trap reactive carbonyl species, including MG, forming stable adducts that mitigate glycation-induced toxicity [26]. PG exhibits anti-inflammatory, anti-apoptotic, and cytoprotective activities in various biological systems [26,27], but its neuroprotective potential under glycation stress remains poorly characterized. Given its ability to neutralize reactive carbonyl species and suppress inflammatory responses, PG may act as a promising compound for preventing metabolic stress-associated neuronal damage.

The phosphatidylinositol 3-kinase (PI3K)/Akt/GSK-3β signaling pathway plays a crucial role in neuronal survival, synaptic plasticity, and cognitive function [28,29]. Activation of PI3K leads to Akt phosphorylation, which in turn inhibits GSK-3β via Ser9 phosphorylation. Evidence suggests that dysregulated PI3K/Akt/GSK-3β signaling results in elevated GSK-3β activity, increased tau hyperphosphorylation and neuronal degeneration [30]. Restoration or activation of PI3K/Akt signaling ameliorates tau hyperphosphorylation and improves cognition in AD-like models [31,32]. However, whether PG can alleviate MG-induced Alzheimer-like cognitive impairment through modulation of the PI3K/Akt/GSK-3β axis remains unknown. Therefore, this study aimed to investigate the neuroprotective effects and underlying mechanisms of PG in an MG-induced mouse model of cognitive dysfunction.

## 2. Results

### 2.1. PG Improves Spatial Learning and Memory in MG-Treated Mice

A brief experimental timeline illustrating MG exposure, PG administration, and behavioral testing is provided in Figure 1A. General physiological parameters of mice are summarized in the Appendix A). Body weights increased steadily throughout the experimental period, with no significant differences among groups (*p* > 0.05). Average food intake and organ weights were also comparable across groups. However, the average daily water intake in the Control group was significantly higher than that in MG- and PG-treated groups (*p* < 0.05), likely because of the unpleasant odor of MG-containing water. Similar reductions in water intake have been reported in mice exposed to MG-containing drinking water [33]. Serum alanine aminotransferase (ALT) levels shown in Appendix A did not differ significantly among groups (*p* > 0.05), indicating that neither 1% MG exposure nor PG administration (up to 100 mg/kg) caused hepatic injury. These findings are consistent with the well-established safety profile of PG, which is classified as “Generally Recognized As Safe” (GRAS) by the U.S. Food and Drug Administration and has been shown to be non-toxic in rodent studies at comparable doses [25]. Spatial learning and memory performance were assessed using the Morris water maze (MWM) test. As shown in Figure 1B, escape latency decreased over the five-day training period in all groups. However, MG-treated mice exhibited significantly longer latencies compared with controls (*p* < 0.05), indicating impaired spatial learning. Similar behavioral impairments have been reported in MG-treated or hippocampal-lesioned rodents, validating the use of this model to assess spatial learning and memory deficits [34,35]. PG supplementation reduced escape latency compared with the MG group, and the high-dose PG (HPG) group showed a clear improvement. On the sixth day of testing, the hidden platform was removed to evaluate memory retention in a spatial probe trial. In the probe trial (Figure 1C), representative swim paths demonstrated that control mice focused their trajectories within the target quadrant, whereas MG-treated mice swam randomly without preference for the platform area. PG treatment restored organized search patterns, particularly in the HPG group, which exhibited concentrated swimming paths near the previous platform location. Quantitative analyses (Figure 1D,E) further supported these findings. The MG group spent the shortest time in the target quadrant (26.6 ± 3.5 s), whereas the PG-treated groups spent 37.7 ± 3.7 s (LPG), 40.8 ± 2.5 s (MPG), and 51.3 ± 4.4 s (HPG). Consistently, the MG group exhibited the lowest number of platform crossings (2.1 ± 0.3), while the PG-treated groups showed progressively higher values of 5.4 ± 0.7 (LPG), 6.2 ± 0.9 (MPG), and 8.4 ± 0.8 (HPG). Statistical analysis confirmed that the HPG group spent significantly more time in the target quadrant and crossed the platform area more frequently than the MG group (*p* < 0.05), demonstrating a clear dose-dependent improvement in spatial memory retention. Collectively, these results indicate that chronic MG exposure induces deficits in spatial learning and memory, whereas PG supplementation effectively mitigates MG-induced cognitive impairment, particularly at higher doses.

### 2.2. PG Alleviates Anxiety-like Behavior and Recognition-Memory Deficits

Figure 2 illustrates the behavioral performance of mice in the open field test and novel object recognition test. In the open field test (Figure 2A), mice in the Control group frequently crossed the central zone of the arena, whereas MG-treated mice displayed trajectories concentrated along the periphery, suggesting increased anxiety-like behavior. In contrast, PG-treated mice exhibited a greater tendency to explore the center area, with higher PG doses producing more frequent entries into the central zone. Quantitative analysis (Figure 2B) showed that the MG group spent markedly less time in the central area (36.3 ± 3.5 s) compared with the PG-treated groups, which spent 46.2 ± 4.0, 58.1 ± 5.6, and 69.4 ± 13.3 s for the low (LPG), medium (MPG), and high (HPG) doses, respectively. The HPG group approached the performance of the Control group (73.2 ± 9.1 s), and this increase was statistically significant compared with the MG group (*p* < 0.05). The number of fecal pellets (Figure 2C) also reflected this trend. MG-treated mice produced an average of 4.1 ± 1.7 pellets, while the PG-treated groups showed dose-dependent reductions (3.9 ± 1.6, 2.5 ± 1.0, and 1.3 ± 0.5 for LPG, MPG, and HPG, respectively; *p* < 0.05 vs. MG). These findings indicate that PG mitigates MG-induced anxiety-like behavior in a dose-dependent manner. In the novel object recognition test (Figure 2D), the discrimination ratio was used to assess recognition memory, with 0.5 serving as the threshold (dotted line) between random and preferential exploration. The MG group showed a markedly reduced discrimination ratio (0.34 ± 0.02), indicating a significant decline in the ability to recognize novel objects. In contrast, PG-treated mice exhibited dose-dependent improvements, with discrimination ratios of 0.59 ± 0.01, 0.65 ± 0.01, and 0.73 ± 0.03 for the LPG, MPG, and HPG groups, respectively (*p* < 0.05 vs. MG). These results demonstrate that PG not only reduces anxiety-like behaviors but also enhances recognition memory compromised by MG exposure. Consistent with previous reports, MG-induced neurotoxicity and hippocampal inflammation are known to impair both anxiety regulation and recognition memory [36,37]. Similarly, phenolic-rich compounds have been reported to reduce anxiety-like behavior and improve recognition memory in rodent models, partly through modulation of neurotransmitter systems and suppression of neuroinflammatory signaling [38,39]. Taken together, these findings suggest that PG confers behavioral protection against MG-induced neuronal dysfunction, improving both emotional and cognitive performance.

### 2.3. PG Mitigates Hippocampal Neuronal Damage and Tau Hyperphosphorylation

Hematoxylin and eosin (H&E) staining of hippocampal sections (Figure 3A) revealed distinct morphological alterations among groups. In the CA1 and CA3 regions, neurons in the Control group were densely and orderly arranged, whereas those in the MG group appeared sparsely distributed, disorganized, and exhibited darkly stained, shrunken cell bodies indicative of neuronal degeneration. These histopathological changes are consistent with previous findings showing that MG administration promotes the accumulation of advanced glycation end-products (AGEs) and neuronal loss in the hippocampus [22]. PG supplementation ameliorated these alterations, with neuronal density partially recovering in the LPG group, although some cellular shrinkage remained, while the HPG group exhibited a near-normal cytoarchitecture, suggesting a dose-dependent neuroprotective effect. Given that PG possesses strong MG-trapping capacity, its administration likely reduced local MG concentrations and attenuated AGE-related neuronal injury [26].

### 2.4. PG Reduces Hippocampal p-Tau and Aβ Accumulation

Western blot analysis (Figure 4) showed that MG exposure markedly increased hippocampal levels of p-tau (Ser396) and Aβ compared with the Control group (*p* < 0.05). The ratio of p-tau to total tau was significantly elevated in the MG group, confirming that 1% MG effectively induced tau hyperphosphorylation and Aβ overproduction in the hippocampus. These observations are consistent with previous reports indicating that MG-induced carbonyl stress accelerates both amyloidogenic processing and tau phosphorylation in Alzheimer-like pathology [22,40]. Administration of PG attenuated these effects in a dose-dependent manner. Although mild reductions were observed in the low- and medium-dose groups, a significant decrease in both p-tau and Aβ levels was detected only in the high-dose PG (HPG) group (*p* < 0.05 vs. MG). The protein expression pattern of p-tau paralleled the immunohistochemical observations (Figure 3B), confirming that PG mitigated MG-induced tau hyperphosphorylation. PG supplementation attenuated these pathological changes in a dose-dependent manner. A significant reduction in the p-tau/tau ratio was observed in the medium-dose (MPG) and high-dose (HPG) PG groups (*p* < 0.05 vs. MG), whereas Aβ levels showed a significant decline only in the HPG group (*p* < 0.05 vs. MG). The protein expression pattern of p-tau paralleled the immunohistochemical findings (Figure 3B), which demonstrated marked tau hyperphosphorylation in the MG group and its gradual attenuation following PG treatment. These results suggest that PG mitigates MG-induced tau and Aβ pathology, likely by trapping MG and suppressing the downstream formation of AGEs that promote GSK-3β-mediated tau phosphorylation and amyloidogenic processing [26]. Collectively, PG appears to preserve hippocampal protein homeostasis under MG-induced carbonyl stress.

### 2.5. PG Restores PI3K/Akt/GSK-3β Signaling Disrupted by MG Exposure

The PI3K/Akt/GSK-3β signaling pathway plays a crucial role in maintaining neuronal survival and synaptic plasticity, and its dysregulation has been implicated in the pathogenesis of AD. The downstream kinase GSK-3β promotes tau hyperphosphorylation and enhances β-secretase activity, contributing to Aβ generation and neuronal injury [32,40]. Under normal physiological conditions, activation of PI3K leads to phosphorylation of Akt at Ser473, generating p-Akt, which subsequently phosphorylates GSK-3β at Ser9, thereby inhibiting its activity and preventing excessive tau phosphorylation. In this study, Western blot analysis was used to evaluate the effects of PG on the PI3K/Akt/GSK-3β pathway in the hippocampus of MG-treated mice (Figure 5). MG exposure markedly reduced the expression of PI3K and decreased the phosphorylation of both Akt and GSK-3β, as reflected by lower p-Akt/Akt and p-GSK-3β/GSK-3β ratios, indicating inactivation of the PI3K/Akt pathway and sustained GSK-3β activity. PG supplementation counteracted these changes in a dose-dependent manner. The expression of PI3K increased progressively with PG dosage, accompanied by elevated p-Akt/Akt and p-GSK-3β/GSK-3β ratios. Statistical analysis revealed that the high-dose PG (HPG) group exhibited a significant increase compared with the MG group (*p* < 0.05), while the medium-dose group showed an upward trend without reaching significance. Restoration of Akt phosphorylation was associated with enhanced phosphorylation (inactivation) of GSK-3β, consistent with the observed reduction in tau hyperphosphorylation (Figure 4). Previous studies have demonstrated that accumulation of AGEs caused by MG inhibits Akt activation, leading to persistent GSK-3β activity and neuronal apoptosis [40,41]. In the present study, PG likely restored PI3K/Akt/GSK-3β signaling by trapping MG and reducing AGE formation, thereby reactivating Akt and suppressing GSK-3β-mediated tau pathology. These findings suggest that PG protects against MG-induced neuronal injury through modulation of the PI3K/Akt/GSK-3β pathway.

### 2.6. PG Suppresses MG-Induced Hippocampal Inflammation

Neuroinflammation plays a critical role in the progression of AD and other neurodegenerative disorders. Activation of astrocytes and microglia triggers the release of pro-inflammatory cytokines, among which tumor necrosis factor-α (TNF-α) and interleukin-6 (IL-6) are key mediators of neuronal injury and synaptic dysfunction [42]. In the present study, Western blot analysis was performed to assess the effects of PG on TNF-α and IL-6 expression in the hippocampus of mice with MG-induced neuronal injury (Figure 6). The Control group exhibited the lowest expression levels of both cytokines, whereas the MG group showed marked upregulation of TNF-α and IL-6, indicating that MG exposure elicited a strong neuroinflammatory response. Elevated levels of these cytokines have previously been shown to disrupt blood–brain barrier integrity and exacerbate neuronal degeneration [26]. Following PG treatment, the expression levels of TNF-α and IL-6 were reduced in a dose-dependent manner. Quantitative analysis revealed that the high-dose PG (HPG, 100 mg/kg) group showed a significant decrease compared with the MG group (*p* < 0.05), while the medium- and low-dose groups displayed a downward trend. These findings indicate that PG attenuates MG-induced hippocampal inflammation, likely through its ability to trap MG and suppress the activation of downstream inflammatory signaling cascades, thereby contributing to its overall neuroprotective effect.

## 3. Discussion

In the present study, the protective effect of PG against MG-induced neurological damage in mice was demonstrated, and several conclusions can be drawn from the findings. The administration of 1% MG and PG at doses of 20, 40, and 100 mg/kg did not affect the overall growth or health status of C57BL/6 mice. Serum ALT levels indicated that PG at these doses was not hepatotoxic, consistent with its well-established safety profile as a food additive recognized as “Generally Recognized as Safe” (GRAS) by the U.S. Food and Drug Administration [25].

Behavioral tests revealed that PG supplementation effectively alleviated MG-induced spatial cognitive deficits and improved emotional and exploratory behaviors. In the novel object recognition test, MG-treated mice displayed impaired long-term memory, whereas PG administration significantly improved discrimination performance, suggesting restoration of recognition memory. MG is a reactive dicarbonyl that promotes the formation of AGEs and induces carbonyl and oxidative stress, which contribute to neuronal dysfunction and cognitive decline [41]. Li et al. [43] demonstrated that quercetin can effectively trap MG and suppress AGE formation in food systems, supporting the hypothesis that MG-scavenging activity contributes to reduced MG-related toxicity. Importantly, Cui et al. [26] further revealed that PG not only traps MG efficiently but also forms a stable mono-MG adduct retaining strong antioxidant and anti-carbonyl activities, thereby reducing both oxidative and glycation stress. Multiple lines of evidence indicate that phenolic compounds modulate oxidative stress and neuroinflammation to preserve cognitive function [44,45]. Recent kinetic modeling of MG-flavonoid reactions has also shown that structural features of phenolic compounds, including hydroxylation and conjugation patterns, determine their MG-trapping rate constants [46], further supporting that PG may act through a similar carbonyl-scavenging mechanism.

Histopathological analyses confirmed that MG induced neuronal shrinkage and decreased cell density in the CA1 and CA3 regions of the hippocampus, while PG treatment preserved neuronal integrity and morphology. Immunohistochemical staining showed that PG suppressed MG-induced tau hyperphosphorylation. Consistent with previous findings, MG accelerates tau aggregation through increased GSK-3β activity and AGE-mediated signaling [41,47]. Given the MG-trapping and antioxidant capacities of PG, it likely mitigates these effects by lowering MG levels and reducing oxidative stress in neuronal tissues, in line with prior evidence that PG protects neurons in rodent forebrain ischemia and that antioxidant strategies preserve synaptic function under oxidative stress [27,48,49]. Future studies will include assessments of oxidative stress indicators, such as SOD or MDA, to clarify whether the neuroprotective action of PG also involves modulation of endogenous antioxidant status under MG challenge.

At the molecular level, PG restored the PI3K/Akt/GSK-3β signaling pathway disrupted by MG exposure. The activation of PI3K and phosphorylation of Akt at Ser473 led to the inactivation of GSK-3β via phosphorylation at Ser9, thereby attenuating tau hyperphosphorylation and Aβ production. A recent review by Basha et al. [50] highlighted that citrus-derived phytochemicals can modulate multiple pathological features of neurodegenerative diseases, including activation of the PI3K/Akt/GSK-3β signaling pathway and suppression of neuroinflammation, which aligns with the mechanistic basis of the present study. These results are consistent with previous reports that activation of the PI3K/Akt pathway ameliorates tau pathology and improves cognition in AD-like models [32]. Future studies will examine downstream effectors of the PI3K/Akt pathway, such as mTOR and FOXO, to determine whether PG exerts broader regulatory effects beyond the Akt–GSK-3β axis and to provide a more comprehensive understanding of pathway involvement. PG treatment also reduced the expression of pro-inflammatory cytokines TNF-α and IL-6 in the hippocampus, suggesting that its neuroprotective effect may be partially mediated through suppression of neuroinflammatory processes, consistent with the anti-inflammatory activity of phenolic antioxidants [33,51]. Future investigations including glial activation markers, including astrocytic (GFAP) and microglial (Iba1), will help clarify whether PG-mediated reductions in cytokines translate into cellular-level regulation of neuroinflammatory responses. Although MG-induced pathology in wild-type mice reproduces key Alzheimer-like phenotypes, transgenic models (e.g., 5xFAD or APP/PS1) featuring genetically driven amyloid or tau pathology would further validate whether PG remains effective across broader AD mechanisms. Such validation is planned for follow-up studies as a next step to strengthen translational relevance.

Collectively, these findings indicate that PG exerts neuroprotective effects against MG-induced neuronal injury by modulating both the PI3K/Akt/GSK-3β pathway, reducing tau hyperphosphorylation and Aβ accumulation, and attenuating neuroinflammatory responses in the hippocampus. These results highlight the potential of PG as a preventive or therapeutic candidate for glycation-associated cognitive impairment and Alzheimer-like neurodegeneration.

## 4. Materials and Methods

### 4.1. Chemicals and Reagents

PG, purity ≥ 98% and dimethyl sulfoxide (DMSO) were purchased from Sigma-Aldrich (St. Louis, MO, USA). Anti-tau (Cat. no. ab76128, 1:5000) and anti-phospho-tau (Ser396) (Cat. no. ab109390, 1:4000) antibodies were obtained from Abcam (Cambridge, UK). Anti-PI3K (Cat. no. 05-217, 1:200), anti-Akt (Cat. no. 05-591MG, 1:1000), and anti-rabbit (Cat. no. AP187P, 1:5000) and anti-mouse (Cat. no. 12-349, 1:5000) secondary antibodies were purchased from Merck Millipore (Darmstadt, Germany). The anti-phospho-Akt (Ser473) (Cat. no. 9271, 1:2000) antibody was obtained from Cell Signaling Technology (Danvers, MA, USA), and the anti-β-actin (Cat. no. 81115-1-RR, 1:20,000) antibody was purchased from Proteintech (Rosemont, IL, USA). The Coomassie protein assay kit and bovine serum albumin (BSA) were purchased from Thermo Fisher Scientific (Waltham, MA, USA).

### 4.2. Animals and Experimental Design

All animal procedures were reviewed and approved by the Institutional Animal Care and Use Committee (IACUC) of the National Pingtung University of Science and Technology (approval no. NPUST-108-080). Eight-week-old male C57BL/6J mice were housed in a temperature-controlled and ventilated room under a 12 h light/dark cycle (lights on from 7:00 a.m. to 7:00 p.m.), with temperature maintained at 22 ± 3 °C and relative humidity at 40–60%. All animals had free access to standard chow and drinking water throughout the experiment. A total of 50 mice were randomly assigned to five groups (*n* = 10 per group) and housed in acrylic cages, five animals per cage. The experimental groups were as follows: Control, MG, LPG + MG, MPG + MG, and HPG + MG. Mice in the LPG, MPG, and HPG groups received PG by oral gavage at doses of 20, 40, and 100 mg/kg/d, respectively. The Control group received normal drinking water, whereas the MG and PG-treated groups were supplied with water containing 1% MG. Daily food and water intake were recorded, and body weight was measured weekly during the 9-week treatment period. Following the treatment, mice were subjected to behavioral assessments, including the Morris water maze, open field, and novel object recognition tests. At the end of the experiment, animals were fasted for 14 h and euthanized by decapitation. Blood and brain tissues were immediately collected for biochemical and histological analyses.

### 4.3. Behavioral Assessments

#### 4.3.1. Morris Water Maze Test

The Morris water maze (MWM) test was conducted over six consecutive days and consisted of two phases: a hidden platform trial (days 1–5) and a probe trial (day 6). The apparatus consisted of a circular black pool (120 cm in diameter, 50 cm in height) filled with water (depth 24 cm; temperature 22 ± 1 °C), rendered opaque with skimmed milk powder. An acrylic platform (23 cm in height) was positioned 1 cm below the water surface in the target quadrant (quadrant 4). Distinct visual cues were attached to the walls of each quadrant to facilitate spatial learning. A camera positioned above the pool recorded all swimming trajectories. During the hidden platform phase, each mouse was released sequentially from quadrants 1 to 4 and allowed to swim freely for a maximum of 2 min to locate the hidden platform. Upon reaching the platform, the mouse was allowed to remain for 10 s. If the mouse failed to locate the platform within 2 min, it was gently guided to the platform and allowed to remain there for 15 s. The latency to find the platform was recorded as the escape latency. On the sixth day, the platform was removed to evaluate spatial memory retention in the probe trial. Each mouse was released from quadrant 2 and allowed to swim freely for 2 min. Swimming trajectories, time spent in the target quadrant, and the number of platform crossings were analyzed using EthoVision XT 6.0 tracking software (Noldus Information Technology, Wageningen, The Netherlands).

#### 4.3.2. Open Field Test

The open field test was performed in an opaque white acrylic chamber (40 × 40 × 40 cm) divided into 16 equal squares marked on the floor. A camera was positioned above the apparatus to record behavior. Before each trial, the chamber was cleaned with 75% ethanol and allowed to air-dry for 2 min to remove residual odor. Each mouse was gently placed in a corner of the open field facing the wall and allowed to explore freely for 10 min. After each session, the animal was removed, and the apparatus was cleaned again with 75% ethanol and air-dried for 2 min before introducing the next subject. The number of fecal boli was recorded manually. The duration and movement trajectories in the central and peripheral zones were analyzed using EthoVision XT 6.0 tracking software.

#### 4.3.3. Novel Object Recognition Test

The novel object recognition test was performed over three consecutive days and consisted of three phases: habituation, familiarization, and test. On day 1 (habituation), mice were placed individually in an opaque white acrylic box (40 × 40 × 40 cm) and allowed to explore freely for 10 min. The apparatus was cleaned with 75% ethanol and air-dried for 2 min between sessions to remove residual odor. On day 2 (familiarization), two identical objects were positioned in the upper left and upper right corners of the box, each 10 cm from the adjacent walls. Mice were introduced into the chamber following the same procedure as on day 1 and allowed to explore freely for 10 min. On day 3 (test session), one of the familiar objects was replaced with a novel object of different shape and texture. A camera positioned above the apparatus recorded the 10 min exploration period. The time spent exploring each object was measured, and the discrimination ratio was calculated as:$$Discrimination\;ratio=\frac{T_{N}}{T_{N}+T_{F}}$$

*T_N_*: time spent exploring the novel object; *T_F_*: time spent exploring familiar object.

Behavioral tracking and time analysis were performed using EthoVision XT 6.0 software.

### 4.4. Histological and Immunohistochemical Analyses

After sacrifice, brain tissues were carefully removed and immediately fixed in 10% neutral-buffered formalin (NBF). Paraffin embedding and hematoxylin-eosin (H&E) staining were performed by RAPID SCIENCE Co., Ltd. (Taichung, Taiwan) according to standard histological procedures. For immunohistochemical (IHC) analysis, brain sections were prepared from paraffin-embedded tissue blocks fixed in 10% NBF. Antigen retrieval was performed in pH 6.0 citrate buffer for 20 min. The primary antibody used was Recombinant Anti-Tau (phospho S396) antibody [EPR2731] (ab109390, Abcam, Cambridge, UK) at a dilution of 1:4000 for 30 min at room temperature. The staining procedures, including incubation and visualization, were performed by RAPID SCIENCE Co., Ltd. (Taichung, Taiwan).

### 4.5. Western Blot Analysis

Hippocampal tissues were weighed and homogenized in RIPA lysis buffer supplemented with protease and phosphatase inhibitor cocktails (Thermo Fisher Scientific, Waltham, MA, USA). Total protein content was quantified using a bicinchoninic acid assay (Thermo Fisher Scientific). Equivalent amounts of protein were resolved by SDS-polyacrylamide gel electrophoresis and electrotransferred onto PVDF membranes. The membranes were blocked in 5% non-fat milk prepared in tris-buffered saline containing 0.1% Tween-20 (TBST) for 1 h at room temperature, followed by overnight incubation at 4 °C with the appropriate primary antibodies. After washing, membranes were incubated with HRP-conjugated secondary antibodies for 1 h at room temperature. Protein bands were visualized using an enhanced chemiluminescence (ECL) detection reagent and captured with a Luminescence Image System (Model M2-8068, Hansor, Taichung, Taiwan). Band intensities were quantified using ImageJ software (version 1.44; National Institutes of Health, Bethesda, MD, USA).

### 4.6. Statistical Analysis

Data are presented as mean ± SEM for behavioral assessments (*n* = 10 per group) and mean ± SD for Western blot analyses (*n* = 3 per group). Normality (Shapiro–Wilk) and homogeneity of variance (Levene’s test) were confirmed before applying one-way ANOVA with Tukey’s post hoc test using SPSS Statistics software (version 25.0, IBM Corp., Armonk, NY, USA). Differences were considered statistically significant at *p* < 0.05.

## 5. Conclusions

Overall, our findings show that PG mitigates MG-induced cognitive dysfunction by restoring hippocampal PI3K/Akt signaling, inhibiting GSK-3β activity, and thereby reducing tau hyperphosphorylation and Aβ accumulation. PG also attenuated neuroinflammation, as evidenced by lowered TNF-α and IL-6 expression in the hippocampus. These results support PG as a promising preventive or adjunct therapeutic candidate for glycation-related cognitive impairment and Alzheimer-like pathology. Unlike prior reports focusing mainly on the antioxidant and MG-trapping properties of PG, the present findings provide in vivo evidence of central neuroprotection by demonstrating reduced tau/Aβ pathology and recovery of PI3K/Akt/GSK-3β signaling. This offers new mechanistic insight into the therapeutic potential of PG for glycation-associated neurodegeneration. In addition to the demonstrated neuroprotective effects, several aspects require further clarification: whether PG exhibits comparable efficacy in female mice, whether circulating or brain MG levels reflect treatment responsiveness, and whether prolonged administration influences safety profiles remain to be addressed. These factors will be examined in future studies to strengthen translational relevance and evaluate the long-term applicability of PG as a neuroprotective dietary agent.

## Figures and Tables

**Figure 1 ijms-27-00511-f001:**
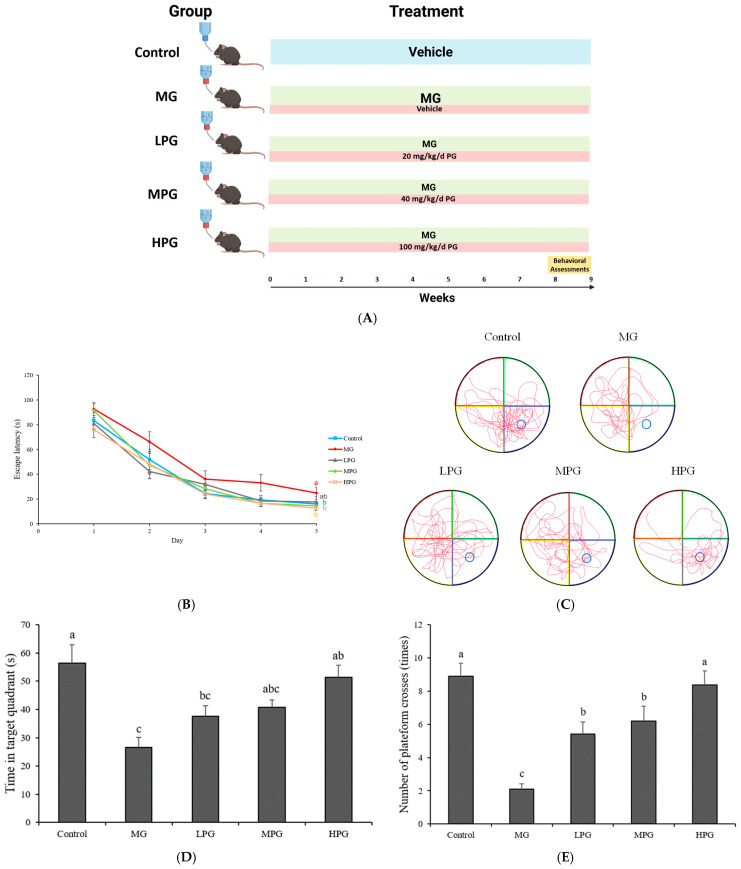
PG improves spatial learning and memory impairment induced by methylglyoxal (MG). (**A**) Mice were orally administered PG (20, 40, or 100 mg/kg) for nine weeks during MG exposure. (**B**) Escape latency during the five-day training sessions of the Morris water maze (MWM) test. (**C**) Representative swim paths in the probe trial conducted on the sixth day after the hidden platform was removed to evaluate spatial memory retention. (**D**) Time spent in the target quadrant and (**E**) number of platform crossings in the probe trial. Values are expressed as mean ± SEM (*n* = 10 per group). Different letters (a–c) indicate significant differences among groups at *p* < 0.05 (one-way ANOVA, Tukey’s post hoc). Control, distilled water (vehicle); MG, 1% MG + vehicle; LPG, MG + 20 mg/kg PG; MPG, MG + 40 mg/kg PG; HPG, MG + 100 mg/kg PG.

**Figure 2 ijms-27-00511-f002:**
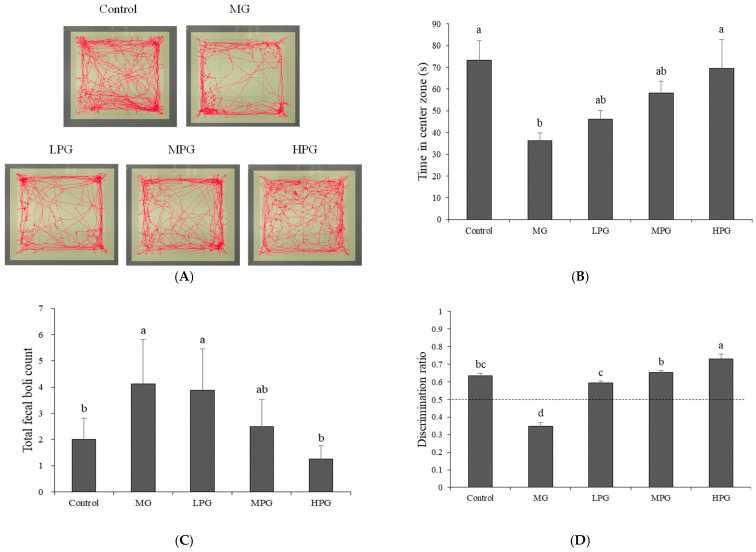
PG attenuates MG-induced anxiety-like behavior and recognition-memory deficits. (**A**) Representative movement trajectories of mice in the open field test. (**B**) Time spent in the central zone and (**C**) total fecal boli count during the open field test, reflecting anxiety-like behavior and emotional reactivity. (**D**) Novel object recognition (NOR) performance showing the discrimination index (the dashed line represents chance performance at a ratio of 0.5). MG exposure induced anxiety-like behavior and reduced recognition memory, whereas PG supplementation significantly improved both parameters in a dose-dependent manner. Values are mean ± SEM (*n* = 10 per group). Different letters (a–d) indicate significant differences at *p* < 0.05 (one-way ANOVA, Tukey’s post hoc). Control, distilled water (vehicle); MG, 1% MG + vehicle; LPG, MG + 20 mg/kg PG; MPG, MG + 40 mg/kg PG; HPG, MG + 100 mg/kg PG.

**Figure 3 ijms-27-00511-f003:**
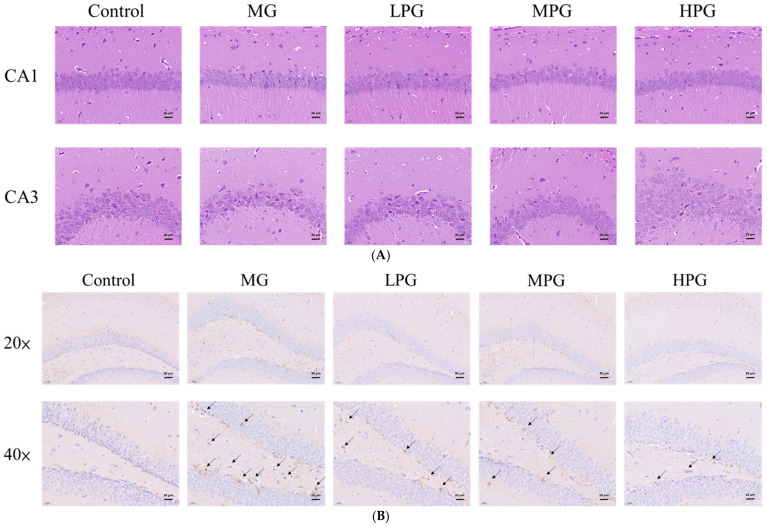
PG mitigates hippocampal neuronal damage and tau hyperphosphorylation. (**A**) Representative hematoxylin and eosin (H&E)-stained hippocampal sections showing the CA1 and CA3 regions. (**B**) Immunohistochemical (IHC) staining of phosphorylated tau (p-tau, Ser396) in the hippocampus (arrows indicate areas with strong immunoreactivity). MG exposure caused neuronal shrinkage and increased p-tau immunoreactivity, whereas PG supplementation preserved neuronal morphology and reduced tau phosphorylation in a dose-dependent manner. Scale bars: 20 μm for H&E; 50 μm (20×) and 20 μm (40×) for IHC images. Control, distilled water (vehicle); MG, 1% MG + vehicle; LPG, MG + 20 mg/kg PG; MPG, MG + 40 mg/kg PG; HPG, MG + 100 mg/kg PG.

**Figure 4 ijms-27-00511-f004:**
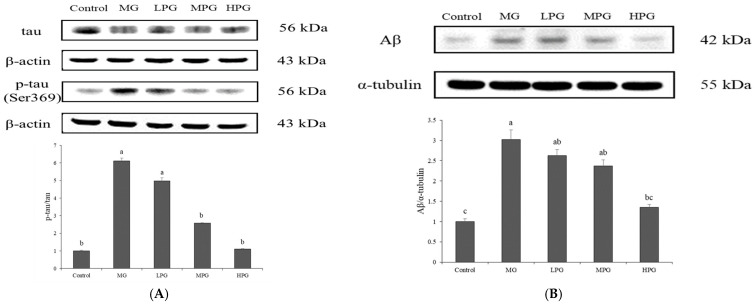
PG reduces hippocampal levels of tau and Aβ proteins. Western blot analysis of (**A**) phosphorylated tau (p-tau, Ser396) and (**B**) Aβ protein in hippocampal tissue. Densitometric quantification normalized to β-actin and α-tubulin, respectively, is shown below each blot. MG treatment markedly increased p-tau and Aβ expression, whereas PG administration significantly reduced both proteins in a dose-dependent manner. Values are expressed as mean ± SD (*n* = 3). Different letters (a–c) indicate significant differences at *p* < 0.05 (one-way ANOVA, Tukey’s post hoc). Control, distilled water (vehicle); MG, 1% MG + vehicle; LPG, MG + 20 mg/kg PG; MPG, MG + 40 mg/kg PG; HPG, MG + 100 mg/kg PG.

**Figure 5 ijms-27-00511-f005:**
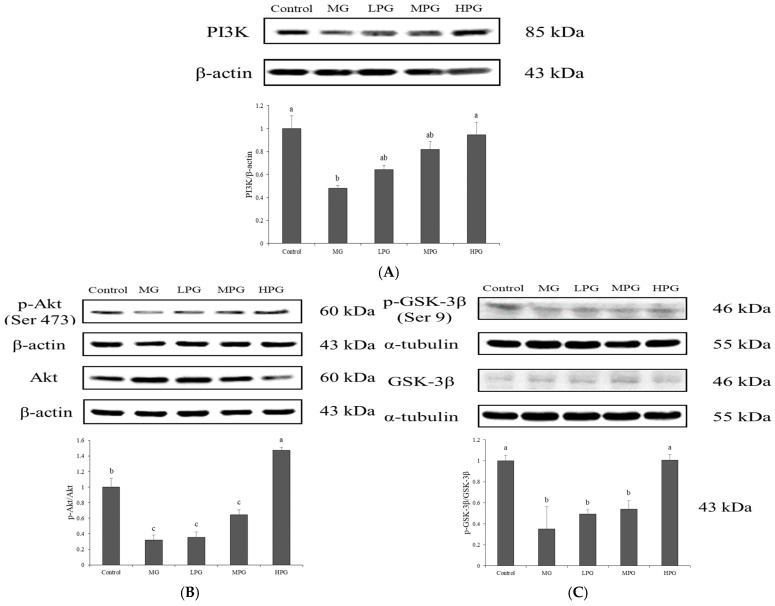
PG restores PI3K/Akt/GSK-3β signaling disrupted by MG exposure. Western blot analysis of hippocampal (**A**) PI3K, (**B**) phosphorylated Akt (p-Akt, Ser473) and total Akt, and (**C**) phosphorylated GSK-3β (p-GSK-3β, Ser9) and total GSK-3β. Densitometric quantification of phosphorylation levels (p-Akt/Akt and p-GSK-3β/GSK-3β ratios) is shown below each blot. MG exposure suppressed Akt and GSK-3β phosphorylation, whereas PG supplementation dose-dependently restored activation of this signaling cascade. Values are expressed as mean ± SD (*n* = 3). Bars with different letters (a–c) differ significantly at *p* < 0.05 (one-way ANOVA, Tukey’s post hoc). Control, distilled water (vehicle); MG, 1% MG + vehicle; LPG, MG + 20 mg/kg PG; MPG, MG + 40 mg/kg PG; HPG, MG + 100 mg/kg PG.

**Figure 6 ijms-27-00511-f006:**
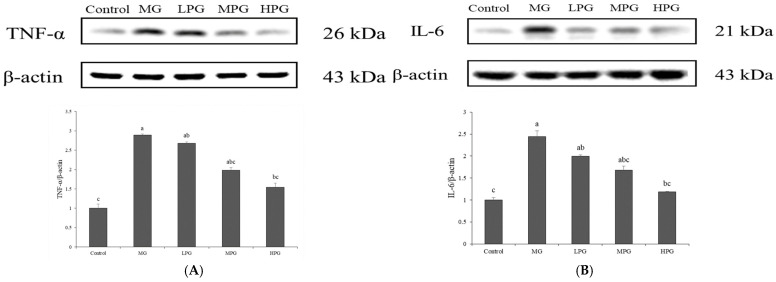
PG suppresses MG-induced hippocampal neuroinflammation. Western blot analysis of (**A**) TNF-α and (**B**) IL-6 protein expression in hippocampal tissue. MG exposure markedly elevated both cytokines, while PG treatment significantly reduced their expression in a dose-dependent manner. Values are expressed as mean ± SD (*n* = 3). Bars with different letters (a–c) differ significantly at *p* < 0.05 (one-way ANOVA, Tukey’s post hoc). Control, distilled water (vehicle); MG, 1% MG + vehicle; LPG, MG + 20 mg/kg PG; MPG, MG + 40 mg/kg PG; HPG, MG + 100 mg/kg PG.

## Data Availability

The raw data supporting the conclusions of this article will be made available by the authors on request.

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
