# Peer review of "Propyl Gallate Attenuates Methylglyoxal-Induced Alzheimer-like Cognitive Deficits and Neuroinflammation in Mice"

_ijms, 2026, doi:10.3390/ijms27010511_

Round 1

Reviewer 1 Report

Comments and Suggestions for Authors

Tsai et al. investigated the protective effects of propyl gallate (PG) against methylglyoxal (MG)-induced Alzheimer-like cognitive impairment in mice using behavioral assays, histological analysis, and molecular studies. The data showed that PG improves cognitive performance, reduces tau hyperphosphorylation and Aβ accumulation, suppresses neuroinflammation, and partially restores PI3K/Akt/GSK-3β signaling in the hippocampus. Overall, the manuscript demonstrated the neuroprotective function of PG for metabolic stress-associated neuro-33 degenerative disorders. However, several major concerns should be addressed before publication.

  1. The authors used Eight-week-old male C57BL/6J in the manuscript, so, how can they detect the tau or Aβ in the WT mice? It would suggested that they should explore the effects of PG in 5XFAD mice or other transgenic mice strains.
  2. While the authors found PG could decrease the expression of tumor necrosis factor-α (TNF-α) and interleukin-286 6 (IL-6) in hippocampus, therefore, I wondering does PG attenuate the overactivation of microglia and astrocytes?
  3. The statistical analysis in figure legends should be modified to enhance the readability of it rather than only using abc.
  4. Providing an experimental timeline would facilitate readers’ understanding of the study design.
  5. Specify the Catalog NO. of antibodies the corresponding dilution ratios used in the manuscript.
  6. The manuscript contains some minor typographical and formatting issues. For example, section numbering and Statistical analysi.

Reviewer 2 Report

Comments and Suggestions for Authors

Dear Authors,

I attach a review of the article „Propyl Gallate Attenuates Methylglyoxal-Induced Alzheimer-Like Cognitive Deficits and Neuroinflammation in Mice”.

This study aimed to investigate the neuroprotective effects and underlying mechanisms of PG in an MG-induced mouse model of cognitive dysfunction. The goals set in the article were achieved. The studies are very interesting and the authors answered the question whether PG can alleviate MG-induced Alzheimer-like cognitive impairment through modulation of the PI3K/Akt/GSK-3β axis.

The manuscript is well prepared in terms of content and editing. However, before publication, the manuscript requires corrections.

Comments:

Line 105-108: …As shown in Figure 1A, escape latency decreased over the five-day training period in all groups; however, MG-treated mice exhibited markedly longer latencies compared with controls, indicating impaired spatial learning…

Rev: One sentence. This problem requires more attention and statistical analysis and a broader description.

Differences should be confirmed by statistical analyses. I propose to do on-way ANOVA for each day, and Repeated Measures ANOVA for each treatment; Tukey test as post-hoc test.

Figure 1: Figures are technically narrowed – Fig 1A,C,D. Should be corrected.

Figure 2: Figures are technically narrowed – Fig 2B,C,D. Signatures (C) (D) should be together with the figures on the page. Should be corrected.

Figure 4: Figures are technically stretched (upper part). Down part (graphs) are illegible. Should be corrected.

Figure 5, 6: See comments above.

Line 460: …3.5. Statistical analysi…

Rev: ?

Reviewer 3 Report

Comments and Suggestions for Authors

I reviewed the manuscript entitled Propyl Gallate Attenuates Methylglyoxal-Induced Alzheimer-Like Cognitive Deficits and Neuroinflammation in Mice.

I agree to accept this manuscript after major revision. 

1) Abstract, 100 mg/kg/day, day should change to d. Please use International System of Units (SI units) instead of spelled-out words, and check and correct similar issues throughout the text. amyloid-β, TNF-α, all Greek letters need to be italicized, check and modify the entire text. amyloid-β (aβ) should change to amyloid-β (Aβ). This is the standard abbreviation.

2)  Keywords, propyl gallate should change to Propyl gallate. Because it is the first keyword, the first letter of its first word needs to be capitalized.

3) which use glutathione (GSH) as a cofactor [19]. Reduced GSH levels impair MG detoxification, GSH only appears twice, there is no need to use abbreviations. Abbreviations are only necessary if they appear three or more times. Otherwise, too many abbreviations will confuse readers. Check and modify similar issues throughout the entire text.   

4) PG exhibits anti-inflammatory, anti-apoptotic, and cytoprotective activities in various biological systems, this statement requires citation(s) from the literature to support the author's argument.

5) 2.1. PG improves spatial learning and memory in MG-treated mice。 The initial letter of each content word in a Level 2 heading should be capitalized. Please check for and correct similar issues throughout the text.

6) 3.1. Chemicals and reagents, Propyl gallate (PG), this abbreviation should be defined in this manner upon its first occurrence, after which the abbreviation may be used directly.

7) temperature 22 ± 1 oC, There should be spaces between numbers and units, except for Celsius and %. Therefore, 22 ± 1 oC should change to 22 ± 1oC.

8) 3.5. Statistical analysi should change to 3.5. Statistical analysis. Did the statistical methods test for data normality and homogeneity of variance? Should it specify whether parametric or non-parametric tests were used?

9) The conclusion section should also address the limitations or shortcomings of the current study. Such as the exclusive use of male mice, the absence of MG level measurements, and the lack of long-term toxicity testing.

10) The description of the mechanism in the abstract is insufficiently clear. While it mentions that PG partially restored the PI3K/Akt/GSK-3β pathway, it does not specify which molecules were up-regulated or down-regulated. Supplementing this information is recommended.

11) The rationale for selecting the MG concentration. Why was 1% MG in drinking water chosen? Is there any literature supporting that this concentration can model AD-like pathology?

12) Is there prior research supporting the selection of PG doses? Were the dose gradients of 20, 40, and 100 mg/kg established based on preliminary efficacy or toxicity studies?

13) Was swimming speed variation controlled for in the Morris water maze test? Are there data showing no differences in swimming speed among groups to exclude the influence of motor ability on the results?

14) What is the magnification of the H&E images in Figure 3A? It is recommended to indicate this in the figure.

15) Could PG potentially affect the intake or metabolism of MG? Were changes in MG levels in serum or brain tissue measured?

16) Given that PG is emphasized as an antioxidant, should oxidative markers such as SOD and MDA have been measured?

17) Were glial cell markers examined for neuroinflammation, such as GFAP (astrocytes) and Iba1 (microglia)?

18) Should downstream effector molecules of the PI3K/Akt pathway, such as mTOR and FOXO, have been examined to more comprehensively elucidate pathway activity?

19) Was gender difference considered? Given that only male mice were used, has this limitation been addressed in the discussion?

20) The potential side effects or long-term safety of PG, particularly at high doses (100 mg/kg), should be discussed.

21) As a common food additive, have similar reports been published regarding the mechanisms by which PG exerts anti-AD-like pathological effects? What breakthroughs does this study offer in terms of mechanistic depth or model specificity? The novelty of this research should be adequately highlighted.

22) This study demonstrates that propyl gallate (PG), a common food antioxidant, alleviates methylglyoxal (MG)-induced Alzheimer-like cognitive deficits in mice. Oral administration of PG improved spatial learning, recognition memory, and anxiety-like behaviors, while reducing hippocampal neuronal damage, tau hyperphosphorylation, amyloid-β accumulation, and neuroinflammation. PG also partially restored the impaired PI3K/Akt/GSK-3β signaling pathway. These findings suggest PG protects against MG-related cognitive decline through modulation of neuroinflammation and survival signaling, supporting its potential as a neuroprotective dietary agent in metabolic stress-associated neurodegeneration.

Round 2

Reviewer 1 Report

Comments and Suggestions for Authors

No more questions.

Reviewer 3 Report

Comments and Suggestions for Authors

The author has made the modifications as requested and addressed my concerns, therefore I agree to accept it in its current form.